# A Fusion Tracking Algorithm for Electro-Optical Theodolite Based on the Three-State Transition Model

**DOI:** 10.3390/s24175847

**Published:** 2024-09-09

**Authors:** Shixue Zhang, Houfeng Wang, Liduo Song, Hongwen Li, Shuai Liu

**Affiliations:** 1Changchun Institute of Optics, Fine Mechanics and Physics, Chinese Academy of Sciences, Changchun 130033, China; zhangsx@ciomp.ac.cn (S.Z.);; 2University of Chinese Academy of Sciences, Beijing 100049, China

**Keywords:** range measurement and control, photoelectric theodolite, data fusion, fusion tracking

## Abstract

This study presents a novel approach to address the autonomous stable tracking issue in electro-optical theodolite operating in closed-loop mode. The proposed methodology includes a multi-sensor adaptive weighted fusion algorithm and a fusion tracking algorithm based on a three-state transition model. A refined recursive formula for error covariance estimation is developed by integrating attenuation factors and least squares extrapolation. This formula is employed to formulate a multi-sensor weighted fusion algorithm that utilizes error covariance estimation. By assigning weighted coefficients to calculate the residual of the newly introduced error term and defining the sensor’s unique states based on these coefficients, a fusion tracking algorithm grounded on the three-state transition model is introduced. In cases of interference or sensor failure, the algorithm either computes the weighted fusion value of the multi-sensor measurement or triggers autonomous sensor switching to ensure the autonomous and stable measurement of the theodolite. Experimental results indicate that when a specific sensor is affected by interference or the off-target amount cannot be extracted, the algorithm can swiftly switch to an alternative sensor. This capability facilitates the precise and consistent generation of data, thereby ensuring the stable operation of the tracking system. Furthermore, the algorithm demonstrates robustness across various measurement scenarios.

## 1. Introduction

The electro-optical theodolite is an optical device employed for angle measurement through optical principles. Its main purpose is to combine a precise optical system with high-precision sensors to record images of rapidly moving objects such as missiles and rockets. Furthermore, it is designed to determine various parameters, including the trajectory, the velocity, and orientation of these objects [1,2,3].

To enhance its detection capabilities, a singular electro-optical theodolite is integrated with cameras exhibiting diverse characteristics, such as distinct operational wavelengths and focal lengths. These typically encompass visible light cameras, near-infrared cameras, mid-wave infrared cameras, and long-wave infrared cameras [4]. A single sensor is limited in its ability to observe targets within a specific range and is vulnerable to environmental disturbances and fluctuations in detector imaging quality. These constraints may lead to inaccuracies in representing the true state of the target or even result in tracking failures [5]. For instance, visible light sensors offer high resolution but require adequate lighting conditions, while infrared sensors can effectively address target obstruction caused by clouds and fog but provide lower spatial resolution. By capitalizing on the complementary features of different sensors, it is possible to reduce uncertainties associated with data from a single sensor to achieve consistent observations of the tracked target [6,7,8]. Multisensor data fusion offers several advantages over using a single sensor [9,10].

(1)The enhancement of detection performance is achieved by utilizing multiple measurements of the target concurrently. By employing fusion computation, the precision of target state estimation is increased, thereby reducing the level of uncertainty in the information obtained.(2)The enhancement of the tracking system’s robustness is achieved through the incorporation of multiple sensors, providing redundancy during target field tests to ensure the detection of the target by at least one sensor. This redundancy serves to facilitate the smooth progression of the mission.

In the context of an electro-optical theodolite, the objective of state estimation during automatic tracking mode is to achieve the most objective and precise value attainable. Likewise, the aim of sensor fusion is to ascertain a measurement angle deemed the most reliable among all sensors, thereby enhancing the servo system’s ability to accurately track the target’s position. Consequently, the data fusion process employed in an electro-optical theodolite is classified as data-level weighted fusion.

Differences in sensor resolution are evident across different camera types, particularly between visible and infrared sensors. This disparity leads to varying fields of view per pixel. As a result, aggregating pixel values related to target deviation from multiple sensors lacks meaningful interpretation. Instead, the synthesized angles derived from each sensor can be utilized as distinct observational inputs for fusion computations. In cases where a sensor encounters a deviation in its observation angle, the weighted fusion algorithm should promptly adjust by reducing the weight attributed to data from that sensor. This approach ensures the precision of the fusion result, demonstrating a high degree of fault tolerance and robustness in the fusion algorithm with respect to observations [11].

During the data fusion process, the conventional approach of assigning equal weight to measurement components through the arithmetic mean overlooks differences in measurement accuracy, indicating a lack of flexibility. The adaptive weighted fusion algorithm has gained widespread application in diverse fields owing to its advantages, including cost-effectiveness in engineering implementation and reduced dependence on prior information. Scholars have introduced several algorithms aimed at determining appropriate weight values in this context.

Reference [12] discusses the determination of the optimal weighting factor by minimizing the mean square error using a recursive approach to estimate variance, thereby implementing an adaptive weighted fusion algorithm. Reference [13] explores the challenge of multi-sensor fusion in drone systems and proposes a methodology that combines neural networks with extended Kalman filters to precisely predict the drone’s position. Reference [14] presents an adaptive stochastic weighted fusion method tailored for estimating variable targets. This technique leverages the relative fluctuations in collected signals to derive balancing factors that effectively reduce mean square errors. Reference [15] introduces observation support to illustrate the consistency of measurements among a sensor and other sensors. Sensors demonstrating strong support are generally considered more reliable, resulting in higher weights being assigned to them. This process enables adaptive weight adjustments, ultimately enhancing measurement reliability.

In the domain of range measurement and control, reference [11] addresses the challenge of real-time measurement in external ballistics by employing multiple external measurement devices that correspond to various information sources. This approach aims to statistically analyze random measurement errors and facilitate data fusion in real-time. However, the algorithm presented does not indicate any enhancements in the tracking stability of individual measurement devices. Similarly, reference [16], which focuses on continuous tracking measurements using theodolites, also advocates for the fusion of multiple measurement sources. Nonetheless, this paper primarily implements weighted fusion to enhance measurement accuracy, without adequately considering the subsequent application of the fused values derived from a single electro-optical tracking device in the context of autonomous tracking.

Hence, it can be seen that the utilization of adaptive weighted fusion algorithms in optical measuring instruments, like photogrammetric devices within certain domains, is constrained. This limitation is primarily attributed to the isolated nature of existing research on weighted fusion algorithms, which often fails to integrate with the comprehensive tracking system of photoelectric theodolite [17]. Nonetheless, the incorporation of data integration should strive to enhance the autonomous and reliable tracking functionalities of the system.

In the context of closed-loop automatic tracking systems utilizing multiple sensors, a common approach involves manually selecting sensor data with high image clarity and stable observations for transmission to the onboard servo [18]. If a sensor fails to accurately measure the off-target amount of the target, the data from that sensor is considered invalid. Alternatively, if a sensor can capture the off-target amount but the output is significantly disrupted and diverges from the actual target position, manual intervention is required to select alternative sensors that provide accurate and precise output [19]. The effectiveness of this process relies heavily on the operator’s expertise, presenting challenges in meeting automation standards for equipment operation. Additionally, data obtained from different sensors may have discontinuities, and switching between sensors could potentially affect the consistent operation of the photogrammetric system. Weighted data fusion techniques can assess the reliability of individual sensors, enabling the integration of weighted fusion algorithms with sensor selection for improved practical significance and feasibility.

The optimal utilization of various sensors and the integration of combined data are fundamental goals of fusion tracking algorithms, with the primary aim of ensuring system stability. In instances where the present sensor does not meet the prescribed output criteria, the fusion tracking algorithm proposed in this research is engineered to smoothly switch to an alternate route of sensor observations that offer both accuracy and efficiency. This shift ensures a continuous and reliable stream of measurement output throughout the entirety of the operation.

## 2. Adaptive Weighted Fusion Algorithm

### 2.1. Adaptive Weighted Fusion Algorithm Based on Error Covariance Recursion

The photoelectric theodolite is furnished with dual servo control systems for azimuth and elevation, both characterized by comparable structures. During automatic tracking operations, the image processing subsystem identifies the target’s location within the present frame image, computes the target’s deviation from the center of the field of view (known as the off-target amount), evaluates this deviation, and sends it to the onboard servo. This closed-loop mechanism operates by directing the platform to modify its orientation to ensure the target remains within the camera’s field of view [20].

Because of differences in sensor and optical system characteristics, the off-target amounts obtained from image analysis by individual sensors need to undergo normalization preprocessing [21].
(1)$$\left\{\begin{array}{c} \Delta A=\frac{m_{x}}{f}\cdot \frac{180}{\pi }\cdot \Delta X\\ \Delta E=\frac{m_{y}}{f}\cdot \frac{180}{\pi }\cdot \Delta Y \end{array}\right.$$

The variables ΔX and ΔY denote the off-target amounts in the horizontal and vertical axes, respectively. The symbol f is used to denote the focal length of the respective camera, whereas $m_{x}$ and $m_{y}$ refer to the dimensions of the detector pixels. The azimuth angle A and elevation angle E of the target relative to the observatory are determined by synthesizing ΔA and ΔE with the encoder values $A_{e}$ and $E_{e}$, as outlined in Equation (2).
(2)$$\left\{\begin{array}{l} A=A_{e}+\Delta A/\cos(E_{e})\\ E=E_{e}+\Delta E \end{array}\right.$$

The preprocessing steps, as described in Equations (1) and (2), ensure the effective integration of data. The primary objective of monitoring measurements in the electro-optical theodolite is to consistently acquire target images and document target quantities and encoder values [22]. Consequently, the theodolite establishes the target’s position by gauging the angles between the target and the station. This process implies that the incorporation of angle measurements obtained by the theodolite relates to real-time data fusion at the operational level.

The determination of the optimal weighting coefficients for integrating multiple sensors is presented in Equation (3).
(3)$$w_{i}=\frac{1}{\left|{\sigma_{i}}^{2}\sum_{i=1}^{n}1/{\sigma_{i}}^{2}\right|},i=1,2,\ldots ,n$$
(4)$$\hat{X}=\sum_{i=1}^{n}w_{i}X_{i}$$

X represents the observed object, while the measurements of sensors are indicated as $X_{i}\left(i=1,\ldots ,n\right)$. These measurements are mutually independent and serve as unbiased estimates of X. The variances of the sensors are denoted as ${\sigma_{1}}^{2},{\sigma_{2}}^{2},\ldots ,{\sigma_{n}}^{2}$.

In practical tracking scenarios, the lack of prior knowledge about the angles of the target presents a difficulty when attempting to apply optimal weighting theory based on sensor variances. Additionally, the dynamic nature of moving targets within the tracking field complicates the reliance on instantaneous measurements for determining sensor weights, as this approach fails to account for the continuous evolution of measurement data over time. Consequently, this oversight leads to a less-than-optimal utilization of measurement information in the calculation of sensor weights.

The formula proposed in the citation [23] outlines a recursive computational method for the instantaneous estimation of the error covariance of individual sensors and the concurrent calculation of weighting coefficients, as illustrated in Equation (5).
(5)$${{\hat{\sigma }}_{i}}^{2}\left(k\right)=\frac{k-1}{k}{{\hat{\sigma }}_{i}}^{2}\left(k-1\right)+\frac{k-1}{k^{2}}[X_{i}(k)-X_{f}(k-1){]}^{2}$$

Here, $X_{i}(k)$ represents the measurement value of sensor i at time k, and ${\bar{X}}_{i}(k-1)$ is the mean value of measurements from this sensor up to time k−1.

### 2.2. The Enhanced Adaptive Weighted Fusion Algorithm

Equation (5) assumes accurate real-time measurements obtained from sensors strategically placed around the target state. It incorporates the time-domain mean as a potential measurement value of the target at the current moment. However, if a sensor encounters substantial disruptions, like the gimbal incorrectly recognizing other aerial entities as the target and misjudging the deviation, this assumption may become invalid. In such cases, the time-domain mean becomes inadequate for the iterative calculation of error covariance. To address this challenge, it is recommended to replace ${\bar{X}}_{i}(k-1)$ with the latest fusion value as specified in Equation (6).
(6)$${{\hat{\sigma }}_{i}}^{2}\left(k\right)=\frac{k-1}{k}{{\hat{\sigma }}_{i}}^{2}\left(k-1\right)+\frac{k-1}{k^{2}}[X_{i}(k)-X_{f}(k-1){]}^{2}$$

On the other hand, as the sampling time gradually increases, the limit in Equation (7) holds true.
(7)$$\underset{k\to \infty }{lim}\frac{k-1}{k}=1,\underset{k\to \infty }{lim}\frac{k-1}{k^{2}}=0$$

Over time, the influence of historical data error covariance increases, leading to a situation where new errors are largely overlooked. This phenomenon can impede the recursive process, limiting its capacity to promptly adjust to fluctuations in present data. In the event of a sensor anomaly during this period, the expected reduction in associated weight may not occur as anticipated, potentially jeopardizing the effectiveness of the weighted fusion algorithm.

The study suggested in Reference [24] advocates for the utilization of the least squares extrapolation technique to estimate the current angle measurement for improving the precision of error covariance recursion. Introducing a decay factor denoted as α in the recursion formula serves to regulate the balance between historical data error covariance and current data, irrespective of the sampling interval.

The extrapolated value for the current time step, denoted as $X_{p}\left(k\right)$, is obtained through the utilization of fusion values ranging from $k-N_{fit}$ to k−1, encompassing $N_{fit}$ points, in a least squares approach, as illustrated in Equation (8).
(8)$$X_{p}\left(k\right)=\sum_{t=k-N_{fit}}^{k-1}w\left(t\right)X_{f}\left(t\right)$$

As a result, the improved recursive formula is outlined in Equations (9) and (10).
(9)$${{\hat{\sigma }}_{i}}^{2}\left(k\right)=\frac{k-1}{k}{{\hat{\sigma }}_{i}}^{2}\left(k-1\right)+\frac{k-1}{k^{2}}[X_{i}(k)-X_{p}(k){]}^{2}$$
(10)$${{\hat{\sigma }}_{i}}^{2}\left(k\right)=\alpha {{\hat{\sigma }}_{i}}^{2}\left(k-1\right)+(1-\alpha )(X_{i}(k)-X_{p}(k){)}^{2}$$

At each time step, the recursive solution for ${{\hat{\sigma }}_{i}}^{2}\left(k\right)$ and its substitution into Equation (3) result in the determination of the adaptive weighting coefficients.

The accurate ballistic measurement data is anticipated to exhibit a continuous curve pattern due to the consistent movement of the target. The improved weighted fusion technique, which depends on the real-time extrapolation of historical data, calculates additional error covariance to enhance the detection of significant deviations in sensor data. This method enables timely modifications to the weighting factors.

In the context of the electro-optical theodolite, $X_{i}\left(k\right)$ denotes the azimuth and elevation values obtained through a weighted fusion of a sensor’s off-target amount and encoder data. During the initial phases of algorithm operation, if the number of sample points does not satisfy $N_{fit}$, the fusion value is determined as the arithmetic mean.

## 3. Fusion Tracking Algorithm Based on Three-State Transition Model

### 3.1. The Three-State Transition Model and Fusion Tracking Principle

The electro-optical theodolite is a widely used multi-sensor tracking and measurement system. Its fusion tracking algorithm functions on the premise that there will always be at least a sensor available to detect the target and one to provide a measurement that closely reflects the actual state of the target for tracking. To meet this requirement, the fusion tracking algorithm is programmed to automatically switch between different sensors to ensure that the currently chosen sensor delivers the most accurate measurement of the target’s state.

The algorithm must first establish the priority ranking of sensors, which is closely tied to the sensor type, task requirements, and environmental conditions. For example, visible light sensors are typically preferred for their high resolution and are often selected as the primary sensor due to their superior performance. Following this, short-wave infrared, mid-wave infrared, and long-wave infrared sensors are prioritized in sequence, each contributing to a priority queue. This queue represents the balance between accuracy and stability when multiple sensors are simultaneously providing data. The priority queue remains consistent throughout the algorithm’s operation. In the subsequent discussion, sensors are identified by priority sequence numbers, with $Z_{1}$ representing the highest priority sensor, followed by $Z_{2}$, $Z_{3}$, and so forth up to $Z_{n}$.

The fusion tracking algorithm is characterized by two main components: firstly, it involves identifying the need for sensor switching, which includes assessing potential sensor interference, malfunctions, or risks of malfunction; secondly, it encompasses the selection of the most suitable sensor for the switch. The former serves as the criterion for switching, requiring continuous monitoring of the sensor’s current and neighboring states to prevent frequent switches triggered by random factors. The latter concerns sensor selection, which should be determined based on a predetermined priority sequence. Additionally, the sensor switching process must ensure the real-time stability of the output measurement values. This chapter introduces a fusion tracking algorithm that is developed using the three-state switching model.

The three-state switching model is based on the intermediate outcomes of the adaptive weighted fusion algorithm. Within the process of error covariance recursion, the incremental error covariance serves to indicate the discrepancy in measurement values at the fusion instance, while the sensor weighting coefficients derived from the error covariance signify the inclination towards specific sensors concurrently. These two elements are utilized in conjunction to ascertain whether a sensor should be transitioned or retained as an output. Initially, the time window is defined as encompassing L sampling moments, inclusive of the present one. The azimuth or elevation angle measurement value of sensor i is denoted as $X_{i}\left(t\right)$, and the magnitude of the residual in the incremental error covariance term is denoted as $M_{i}\left(t\right)$. Using the recursive Formula (10) as an illustration, $M_{i}\left(t\right)$ is expressed as delineated in Equation (11).
(11)$$M_{i}\left(t\right)=\left|X_{i}\left(t\right)-X_{p}\left(t\right)\right|,t=k-L+1,k-L+2,\ldots ,k$$

The three-state switching model outlines the possible states of the current sensor in use at a particular time, namely the hold state, fusion state, and switch state. The key factors in determining these states of the sensor are the variables $M_{i}\left(t\right)$ and $w_{i}$, which function within a designated time frame. The model is characterized by the following definition:

Within the current time window, if all $M_{i}(t)$ values of the azimuth or elevation angle measurements of Sensor i surpass a predefined threshold value th1, or if some $M_{i}(t)$ points exceed th1 while all points of *w_i_* remain below a specified threshold th2, it indicates that the sensor data is not suitable for servo control. This situation is denoted as the switch state. When $M_{i}(t)$ values are significantly high within the window, it indicates a substantial deviation of the measurement $X_{i}(t)$ from historical data, making the comparison of weighted coefficients irrelevant. Therefore, further examination of the weight coefficients is necessary only when certain $M_{i}(t)$ values surpass the acceptable range to determine the measurement’s superiority among other sensors. If the weight coefficient is excessively low, the sensor may not be considered suitable for continued use as the output.

The hold state is identified by the situation in which the azimuth and elevation angle readings of sensor i, observed within a defined time frame, simultaneously satisfy the condition that either all or a portion of the $M_{i}(t)$ values are less than th1, and all $w_{i}$ of sampled points surpass a predetermined threshold th2. This state indicates that the sensor reading displays increased consistency over time in comparison to other sensors, showcasing improved alignment with the pattern of measurement data and heightened dependability in measurement results.

The fusion state is defined by specific requirements: within the time window of sensor i, both azimuth and elevation angle measurements must satisfy the condition where either all or a fraction of the $M_{i}(t)$ values are less than th1, and a real subset of the sampled points whose $w_{i}$ exceed a designated threshold th2 exists. Positioned between the switch state and the hold state, the fusion state serves as an intermediate state. th1 and th2 denote the threshold values for the angle and weight in the prior knowledge, respectively.

According to the three-state transition model, the algorithmic procedure for complete fusion tracking can be outlined as follows:

At the beginning of the algorithm, the azimuth and elevation readings from the sensor Z1, which holds the highest priority, are initially employed as the output. The algorithm assesses the system’s condition by taking into account the sensor selected in the previous time frame and then proceeds to execute a series of actions.

(1)In the event that the sensor is in a state of hold, the algorithm assesses whether this particular sensor holds the utmost priority among all sensors also in the hold state. Should this criterion be satisfied, the algorithm then proceeds to derive the measurement values from this sensor for the current time. Conversely, it will switch to the sensor with the highest priority within the hold state and employ its measured values as the output for the current time.(2)In the fusion state, the algorithm employs the adaptive weighted fusion value of azimuth and elevation measurements as the output for the present time.(3)If none of the conditions mentioned above are met, signifying that the sensor is in the switch state, the algorithm proceeds to the sensor with the greatest priority in the hold state. Following this, it employs the measurement obtained from this sensor as the output for the present time.

The aforementioned algorithmic process illustrates that each sensor continuously updates its state, resulting in state transitions, as depicted in Figure 1. When a switching action occurs, the sensor may exist in either a holding state or a switching state, although the initial conditions are not entirely identical. Transitioning from a holding state occurs when a higher-priority sensor has regained stability following prior interference, with its error metrics and weights satisfying specific criteria over a sustained period. This transition can be characterized as the optimal option among several viable alternatives. A switching state signifies that the current sensor is no longer adequate for tracking measurements, thereby necessitating a transition to an alternative selectable sensor.

It is important to highlight that the switching state of a sensor is “transparent” to the measurement output. In other words, the switching state functions as a virtual state, serving solely as a conditional branch in the determination of the current state of the sensor under evaluation. When a sensor is in a switching state, the algorithm continues to output measurements from other sensors that are in the holding state. From an external perspective, the model consistently provides either the measurements from the holding state or the weighted fusion values.

The algorithm methodically prioritizes the sensors by leveraging existing knowledge, which encompasses recognized measurement precision, and utilizes a three-state discrimination model to evaluate the operational condition of each sensor at a specific point in time. Informed by the findings of this assessment, the algorithm produces the current tracking results of the electro-optical theodolite as it observes the target.

### 3.2. Measurement with Effective Bits Fusion Tracking Algorithm

The image processing subsystem transmits the off-target amount from each sensor to the main control computer for further processing, which subsequently forwards the data to the onboard servo. In practice, the communication protocol between the image processing subsystem and the main control typically incorporates a valid bit to indicate the off-target amount. This segment is denoted by a Boolean value, where “true” signifies the acquisition of a valid off-target amount through image analysis, indicating the validity of the measurement. If the value is “false”, it signifies that the image processing system has not successfully acquired the target, thus indicating the absence of any off-target amount. In such instances, the primary controller should consider the possibility that the sensor may not be able to provide information. The fusion tracking algorithm outlined in Section 3.1 operates under the assumption that each sensor transmits its measurement at every time step, thereby maintaining a constant “true” status for the valid bit associated with the off-target amount throughout each time step.

Sensors with valid bits in measurements raise the following issues:

(1) In fusion tracking algorithms that do not account for valid bits, the sampling time k remains constant for all sensors in the error covariance recursion formula. It denotes the series of measured values from the initiation of the algorithm to the present moment. Given that the interpretation of target images by sensors varies with each time step, the valid sequence of bits for measuring off-target amounts becomes a stochastic process. Measurements lacking a “true” bit do not convey measurement information. Hence, when considering error covariance recursion formulas for measurements containing valid bits, the variable k should denote the index of samples within the relevant valid measurement sequence, known as the valid sampling time. See Figure 2.

(2) The three-state transition model relies on the measurement values obtained within a specific time frame to determine the status of the sensor. If a sensor produces invalid measurements within the defined L-point window, it becomes impossible to apply the three-state criteria. Therefore, it is crucial to establish the candidate sensor queue for the current moment as a collection of sensors where all valid bits for the off-target amount are identified as “true” within the designated time frame. This implies that sensors currently labeled with true valid bits may not be included in the candidate sensor queue; nevertheless, the candidate queue must encompass sensors with true valid bits at the current moment. A sensor must fulfill the requirements of being in the candidate queue and meeting the Hold state condition to be chosen as an output.

In the adaptive weighted fusion algorithm, when valid bits are not taken into account, the first sampling point is unable to undergo the covariance error recursion. At this juncture, the fusion value is determined by calculating the average measurement value from each sensor. When the valid bit of off-target amount is determined by a random process, it is possible for a sensor measurement at a specific moment to align with the first valid off-target amount sampling point. This sensor is described as being in the initial valid moment. Sensors at the initial valid moment are unable to employ the recursive form of the weighted fusion algorithm outlined in Section 2.2 for weight calculation. If, at a specific time point, there exist sensors with a valid off-target amount bit set to “true”, encompassing those present at the initial valid time point, the fusion value is determined as the average of the measurements obtained from sensors with true valid bits at that particular time point, as illustrated in Figure 3.

The flowchart of the algorithm is shown in Figure 4.

The enhanced algorithm is grounded in the three-state transition model; however, it has been specifically developed with feasibility considerations to more effectively correspond with practical engineering applications. This section discusses the fusion tracking algorithm, which tackles the challenges of autonomous fusion and switching related to sensors that possess valid bits in their measurements by incorporating concepts such as the candidate queue.

## 4. Experimental Results and Discussion

### 4.1. Experimental Conditions and Parameters

When utilizing a photoelectric theodolite for tracking measurements, the quality of the image significantly affects the tracking accuracy. In practical situations, the dynamic background of the target may affect the image interpretation. The potential factors that may cause disturbances affecting sensor measurements include:(1)The existence of background regions sharing similar characteristics with the target could lead to fluctuating off-target amounts in image analysis, especially when the target trajectory intersects these regions. The instability observed may result in fluctuating states in sensor measurements that correspond to the respective areas.(2)When new moving objects are detected within the sensor’s field of view and are mistakenly identified as the intended target, the sensor will record angles corresponding to the movement path of the incorrect target. Consequently, this may lead to an inability to precisely track the original target.

The experiment utilizes four sensors, and the optimal measurement configurations for the target are outlined below:
In the azimuthal direction, the target’s initial position is defined as 100°, accompanied by an initial angular velocity of 0.05°/s and an angular acceleration of 0.002${}^{\circ}/s^{2}$ for uniform acceleration motion.In the elevation direction, the target’s initial position is established at an angle of 60°, accompanied by an initial angular velocity of 0.03°/s and an angular acceleration of −0.0005°/s for uniform deceleration motion.The simulation lasts for 20 s, utilizing a sensor sampling rate of 100 Hz and an attenuation factor of 0.95.The measurement values of each sensor are acquired by introducing random noise to the ideal values. Sensors 1 to 4 are labeled as Z1 to Z4, respectively. Sensor 1 is characterized by the lowest noise variance and the highest priority. The error levels of Z2, Z3, and Z4 escalate sequentially based on their priority.


The algorithm utilizes a parameter of L=50, while the angle deviation threshold th1 is established at 0.04. It is important to highlight that the threshold th2 represents the weight threshold of the valid sensors during the fusion calculation process. The quantity of valid sensors fluctuates randomly over time, thus th2 is related to the number of operational sensors and is established as Table 1:

We determine the weighted coefficients of the fusion tracking algorithm by applying Equation (5), denoted as Method 1, where $M_{i}\left(t\right)$ is expressed as per Equation (12).
(12)$$M_{i}\left(t\right)=\left|X_{i}\left(t\right)-{\bar{X}}_{i}\left(t-1\right)\right|,t=k-L+1,k-L+2,\ldots ,k$$

The weighted coefficients of the fusion tracking algorithm are determined by Equation (6), denoted as Method 2, where $M_{i}\left(t\right)$ is expressed as per Equation (13).
(13)$$M_{i}\left(t\right)=\left|X_{i}\left(t\right)-X_{f}\left(t-1\right)\right|,t=k-L+1,k-L+2,\ldots ,k$$

The weighted coefficients of the fusion tracking algorithm are determined by Equation (9), denoted as Method 3, in which $N_{fit}$ is set to 10, where $M_{i}\left(t\right)$ is expressed as Equation (14).
(14)$$M_{i}\left(t\right)=\left|X_{i}\left(t\right)-X_{p}\left(t\right)\right|,t=k-L+1,k-L+2,\ldots ,k$$

The number of simulation points is denoted as N. The optimal azimuth angle value at time k is represented as $A_{b}\left(k\right)$, while the fused value obtained from algorithm i is denoted as $A_{fi}\left(k\right)$. The assessment criteria are delineated as follows:

Mean Square Error (MSE) sequence:(15)$$MSE_{i}\left(k\right)=\frac{1}{k}\sum_{j=1}^{k}(A_{fi}(j)-A_{b}(j){)}^{2}$$

Mean Absolute Error (MAE):(16)$$MAE_{i}=\frac{1}{N}\sum_{k=1}^{N}\left|A_{fi}\left(k\right)-A_{b}\left(k\right)\right|$$

The simulation experiment sets Z1 measurement as invalid at 2.5 s, with the off-target amount valid bit set to false. It becomes valid again at 12.5 s. Z2 measurement is set as invalid at 10 s and remains invalid until the end. At 12.5 s, Z1 experiences interference until 15 s, and at 7.5 s, Z2 experiences interference until 10 s, as shown in Figure 5.

Simulation Experiment 1 and Simulation Experiment 2 correspond to the occurrence of the two types of interference mentioned earlier for Z1 and Z2, respectively. The experiments compare the results of Methods 1, 2, and 3 for handling the same segment of measurements and examine the effect of introducing an attenuation factor in the weighted fusion part of the algorithm. The experiment records the sensor selection sequence of the fusion tracking algorithm and uses azimuth angle as an example to show the algorithm’s output values. It calculates the MSE and MAE sequences between the output values and the ideal measurement values and uses the maximum MSE after the interference starts (referred to as MAX_MSE) to represent the error range. The vertical axis of the sensor selection sequence corresponds to sensors Z1 to Z4. The switch delay time is defined as the temporal interval between the onset of interference and the subsequent activation of the sensor switch, which occurs when the algorithm detects interference in Z2 and executes the switch. This metric facilitates an evaluation of the responsiveness of the fusion tracking algorithm to variations in multi-sensor measurements.

### 4.2. Simulation Experiment 1

In Experiment 1, the recorded values of Z1 and Z2 exhibit significant fluctuations within a defined time period, suggesting irregular variations. The perturbed data are obtained by introducing a substantial amount of random noise to the ideal measurement values.

Figure 6 and Figure 7 illustrate the sensor selection sequence and the azimuthal output values of the fusion algorithm in the presence of interference, employing Method 1. In Figure 6, the vertical axis of the sensor selection sequence, denoted by the numbers 1 to 4, corresponds to sensors Z1 through Z4. This configuration facilitates the observation of the timing associated with sensor selection during the process of algorithm switching. In Figure 7, distinct colors are employed to differentiate the real-time measurements obtained from the four sensors, thereby allowing for a comparative analysis of the fusion values alongside the individual sensor measurements throughout the operation of the algorithm.

Figure 8 and Figure 9 depict the sensor selection sequence and the azimuthal output values of the Fusion Algorithm utilizing Method 2.

Figure 10 and Figure 11 depict the sensor selection sequence and the azimuthal output values of the Fusion Algorithm using Method 3.

Here, the output values for the three methods and the switching delay scenarios at the Z2 interference are compared, as presented in Table 2 and Table 3. The indicators MAX_MSE_A and MAE_A denote metrics related to the azimuth dimension, whereas MAX_MSE_E and MAE_E pertain to metrics associated with the elevation dimension.

Upon the initial instance of interference, when sensor Z2 exhibited fluctuations, Method 1 failed to transition. During the second instance of interference, the system failed to detect the interference from Z1 and reverted to Z1 prematurely, as it did not employ an attenuation factor. The utilization of time-domain mean calculation to determine the updated error covariance for measurements exhibiting a specific pattern of variation is deemed insignificant. Method 2 identified the initial interference and successfully reverted to Z1 by implementing an attenuation factor. Despite the delay in the switching action following the cessation of Z1 interference, Table 2 illustrates that the output error is reduced and maintains a relatively high level of accuracy.

In the first interference scenario, Method 2 and Method 3 demonstrated reduced errors in the output of the fusion state compared to other methods. This was attributed to their superior accuracy in the fused values. Moreover, the incorporation of an attenuation factor enabled Method 2 and Method 3 to promptly account for variations in errors during weight calculations. This indicates that the three-state switch, which relies on weight judgment, successfully maintains a balance between the promptness of sensor switching and the precision of output values.

### 4.3. Simulation Experiment 2

In Experiment 2, Z2 and Z1 began tracking another trajectory at the onset of interference, where their measurements did not exhibit sudden changes but gradually deviated from the ideal values and other sensors that were not affected by interference. After the interference, they returned to normal measurement values.

Figure 12 and Figure 13 show the sensor selection sequence and the azimuthal output values of the fusion algorithm using Method 1 when interference occurs.

Figure 14 and Figure 15 illustrate the sensor selection sequence and the azimuthal output values of the Fusion Algorithm employing Method 2.

Figure 16 and Figure 17 illustrate the sensor selection sequence and the azimuthal output values of the Fusion Algorithm employing Method 3.

The output values for the three methods are compared, and the switching delay situation at the Z2 interference is presented in Table 4 and Table 5.

In Experiment 2, there was a slight increase in the switching time of sensors compared to Experiment 1. Nevertheless, the measurement range of the target remains stable at this juncture, ensuring that the target stays within the field of view throughout the tracking process without abrupt changes, thereby rendering the delay time tolerable.

Method 2, employing fused values for error covariance recursion, and Method 3, utilizing fitted values, demonstrate the extent to which historical measurement values are incorporated during error recursion. In Method 3, greater emphasis is placed on measurement data that conforms to historical patterns of change, leading to the generation of more precise fused values. In terms of switching delay time, Method 2 and Method 3 exhibit a similar performance.

In both Experiment 1 and Experiment 2, there were occurrences where Method 2 and Method 3 failed to revert to Z1 following the cessation of interference. This occurred because during that period, the measurements from each valid sensor closely aligned with the fused values, exhibiting nearly identical weights. The output values at this juncture had already achieved a satisfactory level of accuracy, and the weight of Z1 did not surpass the threshold ***th2***. Nonetheless, the failure to revert to a sensor with a higher priority did not impact the precision of the resulting values. Hence, it can be argued that in both interference scenarios, Method 2 and Method 3 are capable of fulfilling the requirement for sensors to autonomously switch selection.

## 5. Conclusions

(1) This paper presents an enhanced multi-sensor adaptive weighted fusion algorithm that addresses the data fusion challenge in photoelectric theodolites. The algorithm incorporates attenuation factors and least squares extrapolation to improve the fusion process. These enhancements effectively modify the weights of sensors that have been disturbed, consequently enhancing the accuracy of the fused values.

(2) This paper addresses the autonomous stable tracking issue in the closed-loop tracking mode of photoelectric theodolites and defines sensor states as holding state, fusion state, and switching state utilizing new error covariance and weighting coefficients. By evaluating the present sensor state, the algorithm decides whether to produce weighted fusion values or switch sensors. This leads to the development of a fusion tracking algorithm that relies on a three-state model for autonomous switching among multiple sensors. Subsequent enhancements to the three-state transition model and fusion tracking algorithm have rendered it suitable for real-world scenarios involving measurements that include valid bits. Simulation experiments illustrate the effective conversion judgment of the three-state transition, and the sensor can switch promptly in the presence of interference or invalid off-target amounts, thus making the theodolite output consistent and precise. The algorithm demonstrates strong robustness in both interference scenarios.

For the future work of this paper, we hope to broaden the application scenarios of the algorithm to improve the algorithm in practical application experiments.

## Figures and Tables

**Figure 1 sensors-24-05847-f001:**
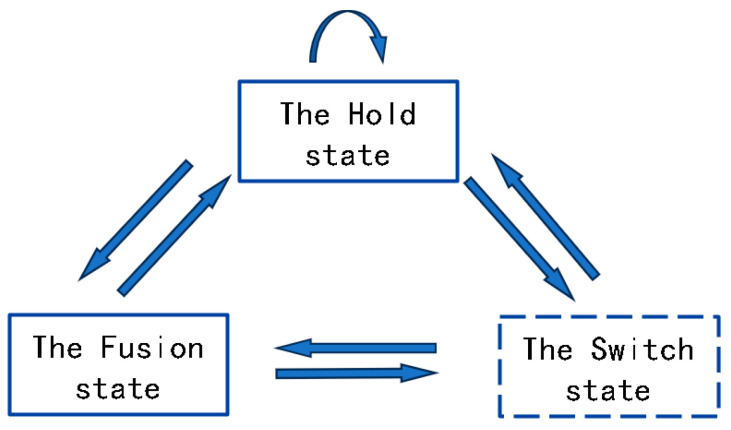
Illustration of state transition.

**Figure 2 sensors-24-05847-f002:**
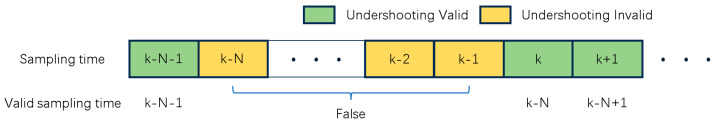
Illustration of effective sampling time.

**Figure 3 sensors-24-05847-f003:**
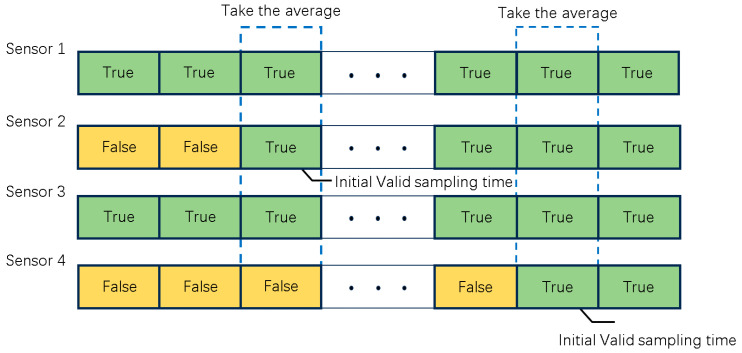
The processing of the sensor at the initial effective time.

**Figure 4 sensors-24-05847-f004:**
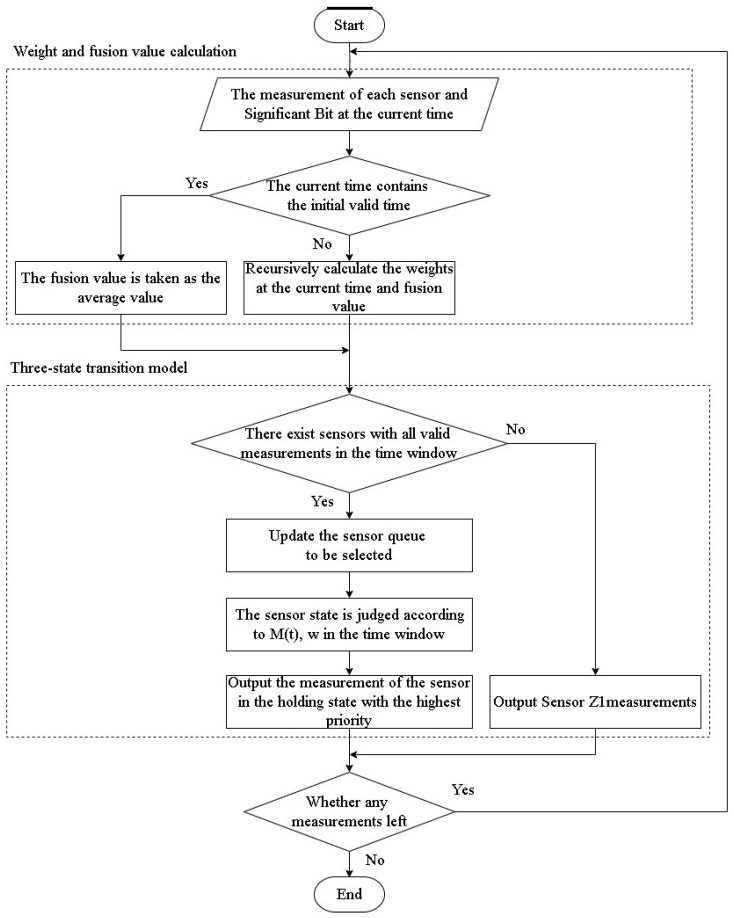
Flowchart for fusion tracking algorithms with valid bits in measure.

**Figure 5 sensors-24-05847-f005:**
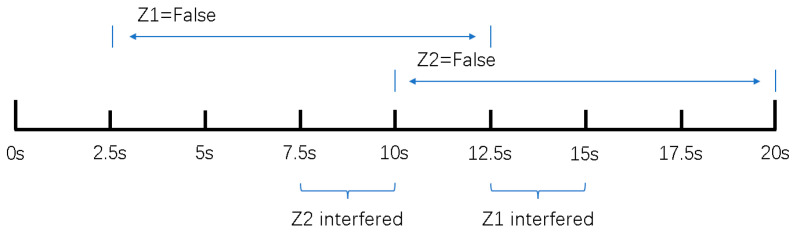
The setting of simulation experiment.

**Figure 6 sensors-24-05847-f006:**
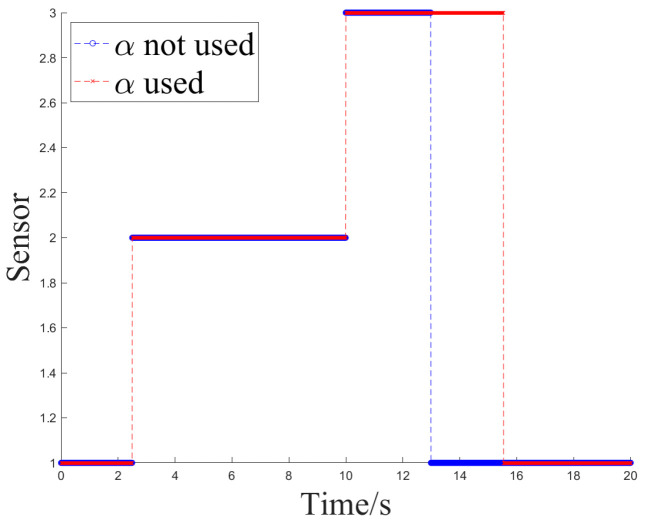
The sensor selection of method 1.

**Figure 7 sensors-24-05847-f007:**
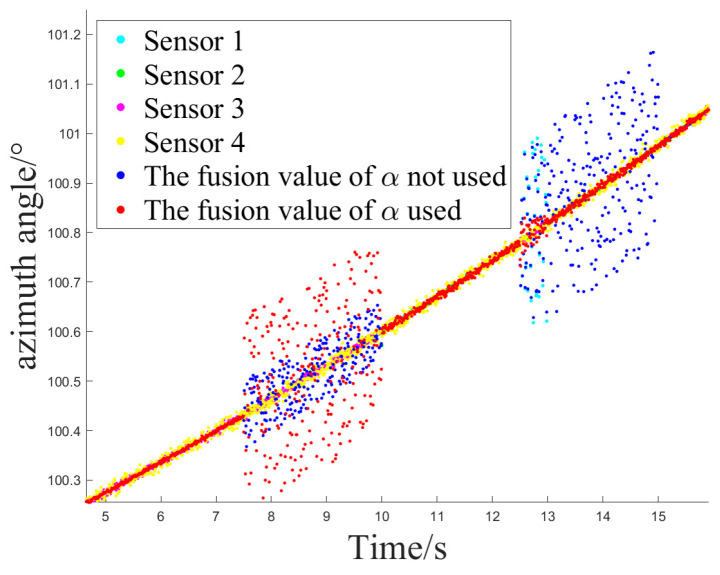
The azimuth output value of method 1.

**Figure 8 sensors-24-05847-f008:**
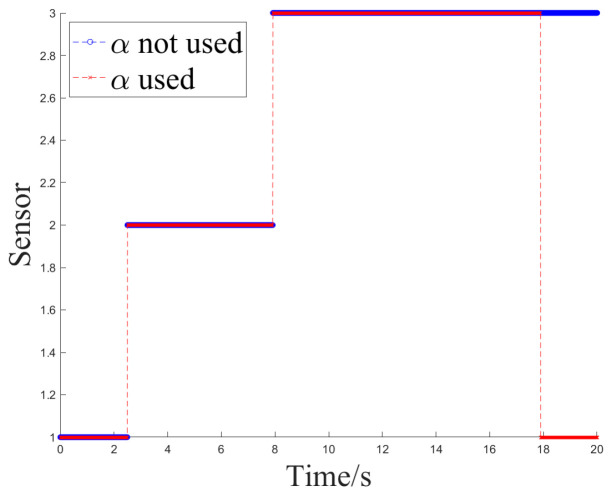
The sensor selection of method 2.

**Figure 9 sensors-24-05847-f009:**
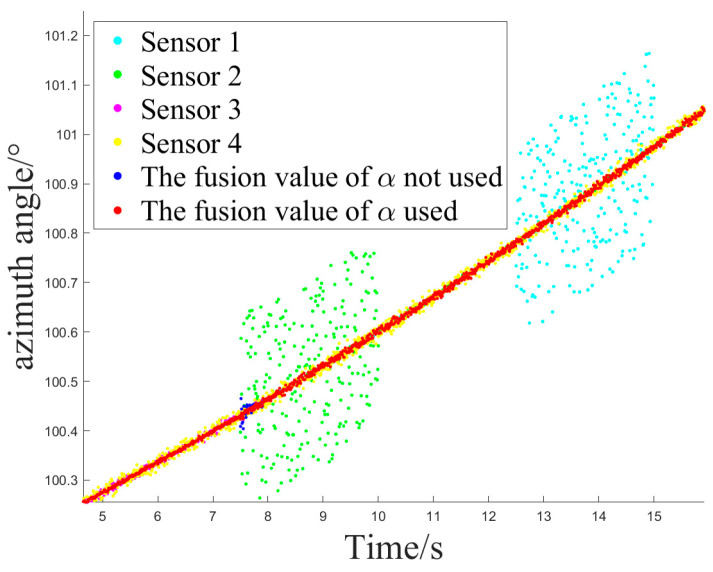
The azimuth output value of method 2.

**Figure 10 sensors-24-05847-f010:**
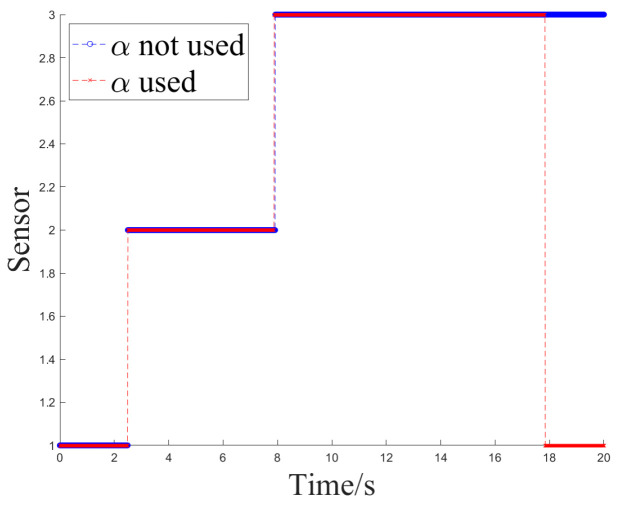
The sensor selection of method 3.

**Figure 11 sensors-24-05847-f011:**
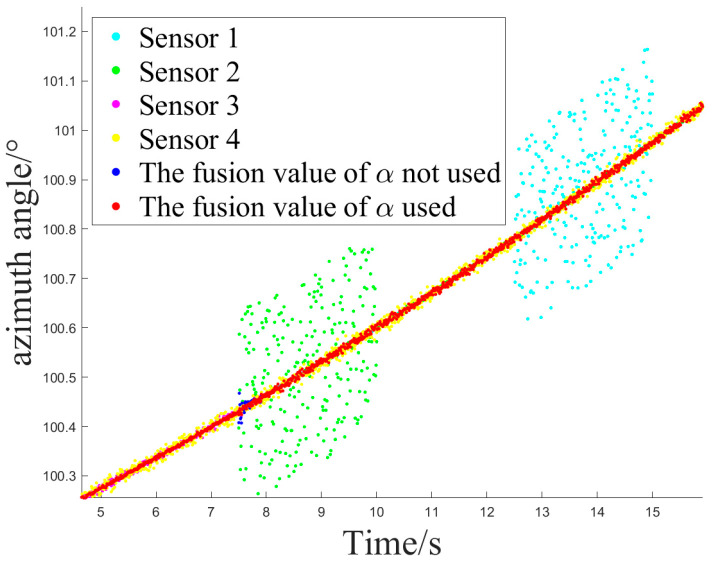
The azimuth output value of method 3.

**Figure 12 sensors-24-05847-f012:**
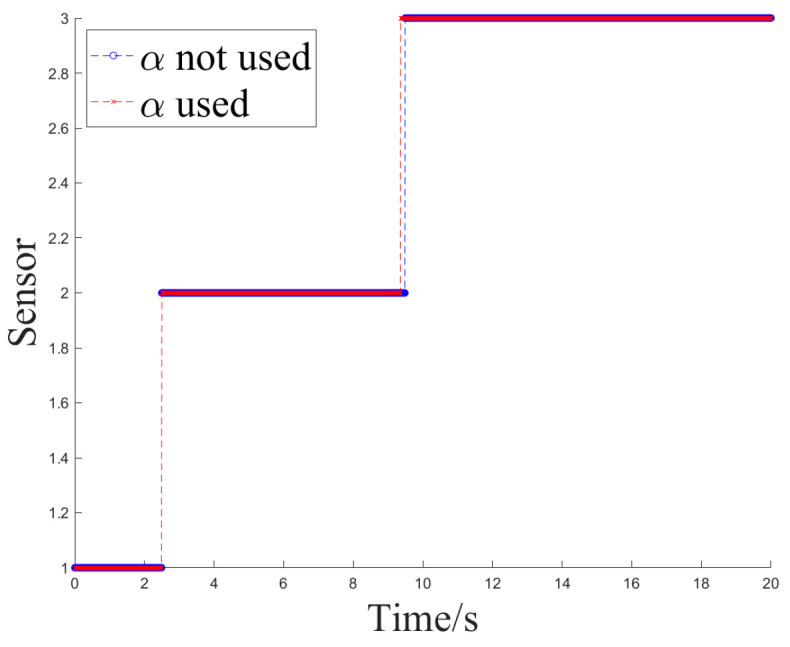
The sensor selection of method 1.

**Figure 13 sensors-24-05847-f013:**
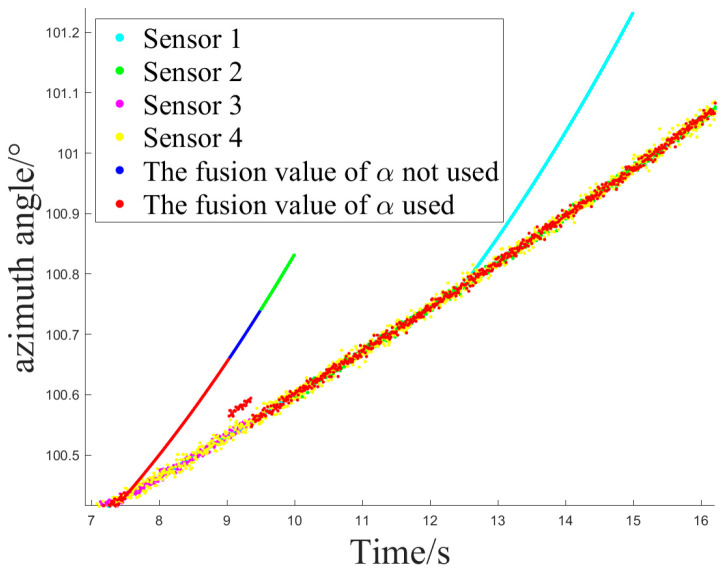
The azimuth output value of method 1.

**Figure 14 sensors-24-05847-f014:**
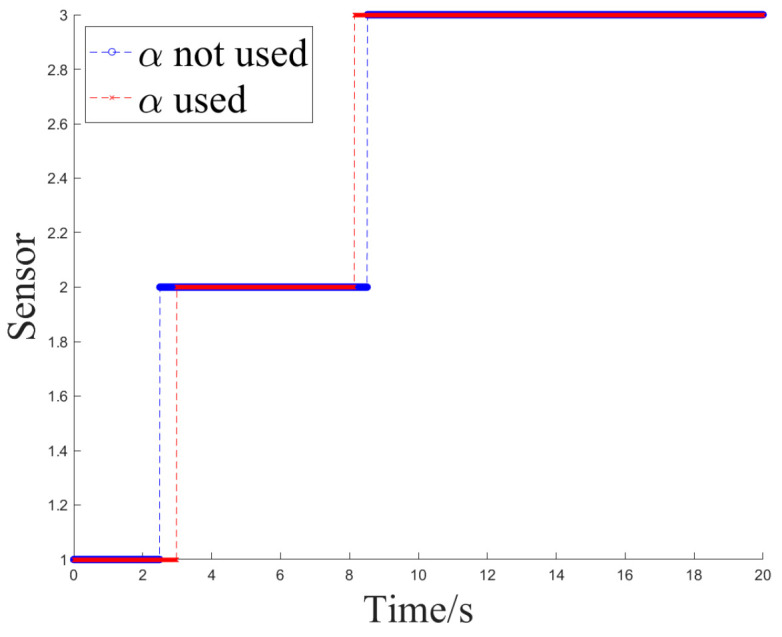
The sensor selection of method 2.

**Figure 15 sensors-24-05847-f015:**
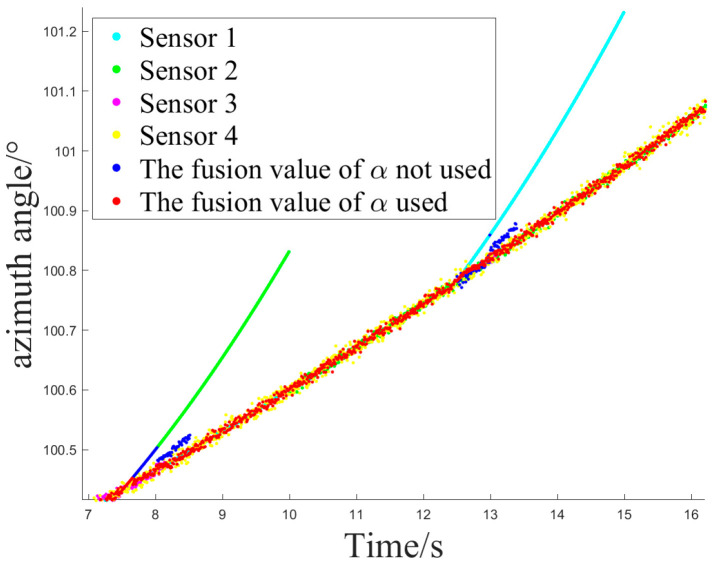
The azimuth output value of method 2.

**Figure 16 sensors-24-05847-f016:**
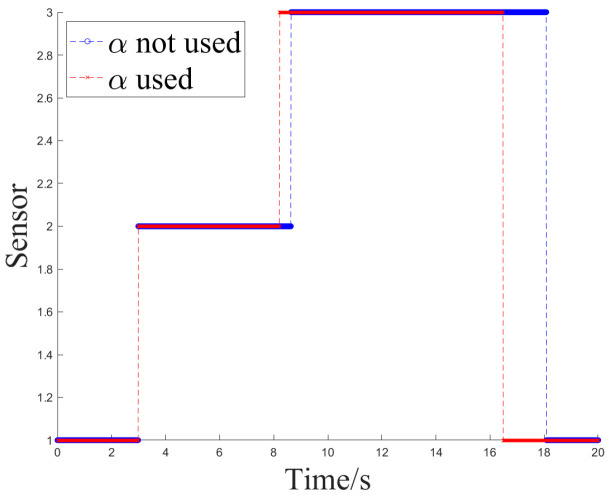
The sensor selection of method 3.

**Figure 17 sensors-24-05847-f017:**
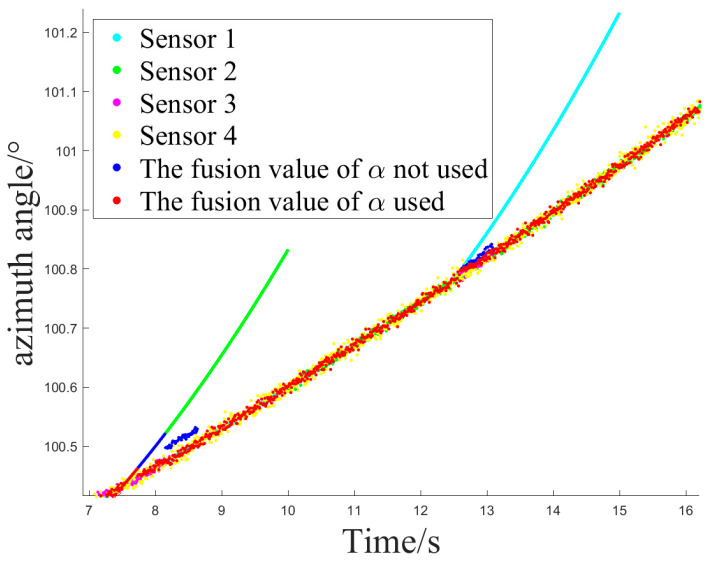
The azimuth output value of method 3.

**Table 1 sensors-24-05847-t001:** The setting for the parameter ***th2***.

Number of Effective Sensors	th2
2	0.35
3	0.25
4	0.1

**Table 2 sensors-24-05847-t002:** Indicators comparison of output values of simulation experiment 1.

	Method 1		Method 2		Method 3	
	Not Using α	Using α	Not Using α	Using α	Not Using α	Using α
MAX_MSE_A (10^−4^)	41.8772	33.1794	0.1337	0.1036	0.1323	0.1034
MAE_A (10^−3^)	25.0461	14.2845	2.6009	2.1840	2.5908	2.1702
MAX_MSE_E (10^−4^)	41.5964	29.5217	0.0059	0.0059	0.0059	0.0058
MAE_E (10^−3^)	24.5366	12.4860	0.5769	0.5702	0.5736	0.5714

**Table 3 sensors-24-05847-t003:** Sensor switching time for simulation experiment 1.

	Method 1		Method 2		Method 3	
	Not Using α	Using α	Not Using α	Using α	Not Using α	Using α
Switching Time (s)	10.01	10.01	7.93	7.93	7.92	7.89
Delay of Switching Time (s)	2.51	2.51	0.43	0.43	0.42	0.39

**Table 4 sensors-24-05847-t004:** Indicator comparison of output values of simulation experiment 2.

	Method 1		Method 2		Method 3	
	Not Using α	Using α	Not Using α	Using α	Not Using α	Using α
MAX_MSE_A (10^−4^)	18.8702	8.4478	0.9550	0.1998	0.6189	0.1907
MAE_A (10^−3^)	10.9451	7.5683	4.5671	3.2638	4.4942	3.2376
MAX_MSE_E (10^−4^)	5.6569	2.2718	0.1343	0.0137	0.0739	0.0114
MAE_E (10^−3^)	4.9615	3.3155	1.1386	0.8025	1.0370	0.7854

**Table 5 sensors-24-05847-t005:** Sensor switching time for simulation experiment 2.

	Method 1		Method 2		Method 3	
	Not Using α	Using α	Not Using α	Using α	Not Using α	Using α
Switching Time (s)	9.51	9.37	8.55	8.21	8.61	8.22
Delay of Switching Time (s)	2.01	1.87	1.05	0.71	1.11	0.72

## Data Availability

Data are contained within the article.

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
