# Peer review of "A Fusion Tracking Algorithm for Electro-Optical Theodolite Based on the Three-State Transition Model"

_sensors, 2024, doi:10.3390/s24175847_

Round 1
Reviewer 1 Report
Comments and Suggestions for Authors
1.The main question addressed by the research is the autonomous stable tracking issue in electro-optical theodolite operating in closed-loop mode.
2.This research is partially improved and optimized based on the research of some researchers, it means that the research of the adaptive weighted fusion algorithm and a fusion tracking algorithm belongs to the original part of this field. Although the algorithm after improved and optimized has proposed a reference scheme for solving the problem of the data fusion challenge in photoelectric theodolites and the autonomous stable tracking issue in the closed-loop tracking mode of photoelectric theodolites. However, specific gaps in this field have not been fully addressed.
3.On the basis of the original part of the field, the innovation of this paper is to propose a multi-sensor adaptive weighted fusion algorithm and a fusion tracking algorithm based on a three-state transition model, which provides the reference direction for the field to solve the problem of photoelectric theodolite data fusion and autonomous stable tracking in the closed-loop tracking mode.
4.In this paper, the author only uses simulation experiments to verify the feasibility of the proposed algorithm improvement and optimization, which has a certain limitation. The feasibility of the proposed algorithm and the robustness mentioned by the author in various test environments still need to be verified in actual environment test experiments
5.The conclusion presented by the author is basically consistent with the evidence and arguments presented. Simulation experiments illustrate the effective conversion judgment of the three-state transition, and the problem of autonomous stable tracking of photoelectric theodolite in closed-loop mode is solved.
6.The references cited in the article are appropriate.
7.There are still some writing problems in this paper should be modified. Some equations are repetitious, such equation 5 and 6. English expressions should be simplified. There are too many words used to demonstrate a simple algorithm. Fig. 3 shows an incomplete picture, the author is advised to change the picture. Figures should be modified, since many curves have been obstructed, such as fig.6, 8,10,12,16.
Comments on the Quality of English LanguageThe English expression in the author's article should be simplified. Statements need to be optimized to improve the overall quality of this article.
Author Response
Comments 1. The main question addressed by the research is the autonomous stable tracking issue in electro-optical theodolite operating in closed-loop mode.
Response 1: Accept. Thanks for your comments and appreciation on the paper.
Comments 2. This research is partially improved and optimized based on the research of some researchers, it means that the research of the adaptive weighted fusion algorithm and a fusion tracking algorithm belongs to the original part of this field. Although the algorithm after improved and optimized has proposed a reference scheme for solving the problem of the data fusion challenge in photoelectric theodolites and the autonomous stable tracking issue in the closed-loop tracking mode of photoelectric theodolites. However, specific gaps in this field have not been fully addressed.
Response 2: Accept. Thanks for your comments and appreciation on the paper.
Comments 3. On the basis of the original part of the field, the innovation of this paper is to propose a multi-sensor adaptive weighted fusion algorithm and a fusion tracking algorithm based on a three-state transition model, which provides the reference direction for the field to solve the problem of photoelectric theodolite data fusion and autonomous stable tracking in the closed-loop tracking mode.
Response 3: Accept. Thanks for your comments and appreciation on the paper.
Comments 4. In this paper, the author only uses simulation experiments to verify the feasibility of the proposed algorithm improvement and optimization, which has a certain limitation. The feasibility of the proposed algorithm and the robustness mentioned by the author in various test environments still need to be verified in actual environment test experiments
Response 4: This paper shows indoor experiments, but various situations are simulated. For the actual environment, we can’t get the equipment data for the reason of secrecy. Basically, the indoor simulation experiment simulated all the actual situations. We add future work part in Chapter 4, the algorithm maybe improved for various application scenarios in the future work. {The modified can be found in line 566-567, page16}.
Comments 5. The conclusion presented by the author is basically consistent with the evidence and arguments presented. Simulation experiments illustrate the effective conversion judgment of the three-state transition, and the problem of autonomous stable tracking of photoelectric theodolite in closed-loop mode is solved.
Response 5: Accept. Thanks for your comments and appreciation on the paper.
Comments 6. The references cited in the article are appropriate.
Response 6: Accept. Thanks for your comments and appreciation on the paper.
Comments 7. There are still some writing problems in this paper should be modified. Some equations are repetitious, such equation 5 and 6. English expressions should be simplified. There are too many words used to demonstrate a simple algorithm. Fig. 3 shows an incomplete picture, the author is advised to change the picture. Figures should be modified, since many curves have been obstructed, such as fig.6, 8,10,12,16.
Response 7: Accept. Some of the writing problems are modified. The corresponding figures (fig.3, fig.6, fig.8, fig.10, fig.12, fig.14, fig.16) mentioned are replaced.
Reviewer 2 Report
Comments and Suggestions for Authors
This paper presents a multi-sensor data fusion algorithm tailored to enhance the accuracy and reliability of opto-electronic warp beam monitoring for fast-moving airborne targets. The proposed method integrates a multi-sensor adaptive weighted fusion algorithm with a fusion tracking algorithm based on a three-state transition model. An enhanced recursive formulation has been developed to estimate error covariance by incorporating attenuation factors and least squares extrapolation, forming the basis for a multi-sensor weighted fusion algorithm. Simulated tests confirm significant enhancements in target tracking accuracy and consistency, particularly in scenarios involving sensor failures or external interference.
(1) In Chapter 1, the literature review provides a sufficient foundation but fails to explore multi-sensor applications beyond those directly related to warp beams. A more comprehensive overview of the research context would strengthen the positioning of this study within the broader technological landscape.
(2) The paper introduces a novel fusion tracking algorithm potentially crucial for the stable tracking of electro-optical warps. However, the discussion on the adaptability of this algorithm across different sensor arrays and environments is limited. Expanding on how the algorithm might be adapted or extended would greatly benefit its practical application.
(3) The number of related research cited is insufficient. It is recommended to incorporate more recent application literature, particularly from 2023-2024, to provide a more current and comprehensive context for the study. For instance, Signal Process of Ultrasonic Guided Wave for Damage Detection of Localized Defects in Plates: From Shallow Learning to Deep Learning; Journal of Data Science and Intelligent Systems.
(4) The comparative analysis with existing state-of-the-art fusion tracking methods is inadequately detailed, leaving the advantages and limitations of the new algorithm unclear. This lack of clarity may undermine the evaluation of the model's improvements or innovations.
(5) To help understanding of how the algorithms function and perform, the inclusion of additional charts and visual elements is recommended. This would provide readers with a more intuitive grasp of the methodologies employed.
(6) Figures 5, 7, 9, 11, and 15 illustrate sensor selection but lack a clear legend or explanation for the color coding representing different sensor states, complicating interpretation.
(7) The mathematical formulas and descriptions of the algorithms may not be accessible to non-specialist readers. These should be simplified or accompanied by clearer explanations to enhance comprehension.
(8) The authors may add more detection articles for applications in various fields ( 3D vision technologies for a self-developed structural external crack damage recognition robot; Automation in Construction. Performance Metrics of an Intrusion Detection System Through Window Based Deep Learning Models; Journal of Data Science and Intelligent Systems.)
Author Response
Comments 1: In Chapter 1, the literature review provides a sufficient foundation but fails to explore multi-sensor applications beyond those directly related to warp beams. A more comprehensive overview of the research context would strengthen the positioning of this study within the broader technological landscape.
Response 1: Accept. We add some explanation in Chapter 1 to describe the discussion scope of this paper. That is the data fusion process employed in an electro-optical theodolite belongs to data-level weighted fusion, Instead of fusion in other levels. We add some references in Chapter 1 to show a more comprehensive overview of the related research. {The modified can be found in line 59-64, page2}.
Comments 2: The paper introduces a novel fusion tracking algorithm potentially crucial for the stable tracking of electro-optical warps. However, the discussion on the adaptability of this algorithm across different sensor arrays and environments is limited. Expanding on how the algorithm might be adapted or extended would greatly benefit its practical application.
Response 2: Accept. We add some description on how the algorithm is adapted to actual experiment. This paper shows indoor experiments, but various situations are simulated. For the actual environment, we can’t get the equipment data for the reason of secrecy. Basically, the indoor simulation experiment simulated all the actual situations. We add future work part in Chapter 4, the algorithm maybe improved for various application scenarios in the future work. {The modified can be found in line 566-567, page16}.
Comments 3: The number of related research cited is insufficient. It is recommended to incorporate more recent application literature, particularly from 2023-2024, to provide a more current and comprehensive context for the study. For instance, Signal Process of Ultrasonic Guided Wave for Damage Detection of Localized Defects in Plates: From Shallow Learning to Deep Learning; Journal of Data Science and Intelligent Systems.
Response 3: Accept. Thank you very much for your suggestion. We have also read 3 recent relevant literature and have benefited greatly from it. Some of the related recent research is incorporated in Chapter 1. The following papers are added to the reference:
- Shang, L., Zhang, Z., Tang, F., Cao, Q., Pan, H., & Lin, Z. Signal Process of Ultrasonic Guided Wave for Damage Detection of Localized Defects in Plates: From Shallow Learning to Deep Learning. Journal of Data Science and Intelligent Systems, 2024, 00(00), 1–16
- Yang D, Du P, Zhong M, et al. A real-time fusion method of external trajectory measurement data based on variable difference method, 2020 IEEE 9th Joint International Information Technology and Artificial Intelligence Conference (ITAIC). IEEE, 2020, 9: 574-577.
- Liu C, Zhao K, Fu F, et al. Real-time data fusion method of multiple photoelectric theodolites based on fault-tolerant mechanism, Second International Conference on Geographic Information and Remote Sensing Technology (GIRST 2023). SPIE, 2023, 12797: 757-764.
Comments 4: The comparative analysis with existing state-of-the-art fusion tracking methods is inadequately detailed, leaving the advantages and limitations of the new algorithm unclear. This lack of clarity may undermine the evaluation of the model's improvements or innovations.
Response 4: Accept. We add more comparative analysis of the existing fusion tracking methods in Chapter 1, and describe the advantage and limitations of them. {The modified can be found in line 100-110, page3}.
Comments 5: To help understanding of how the algorithms function and perform, the inclusion of additional charts and visual elements is recommended. This would provide readers with a more intuitive grasp of the methodologies employed.
Response 5: Accept. We add a new figure and some explanations to describe the algorithm preformation in Chapter 3.1. {The modified can be found in line 297-321, page7}.
Comments 6: Figures 5, 7, 9, 11, and 15 illustrate sensor selection but lack a clear legend or explanation for the color coding representing different sensor states, complicating interpretation.
Response 6: Accept. We add an overall explanation for all the figures in Chapter 4.2. {The modified can be found in line 459-465, page12}.
Comments 7: The mathematical formulas and descriptions of the algorithms may not be accessible to non-specialist readers. These should be simplified or accompanied by clearer explanations to enhance comprehension.
Response 7: Accept. We add some descriptions to the formulas in Chapter 3.1, thus some non-specialist readers may be more clear about the proposed algorithm. {The modified can be found in line 297-321, page7}.
Comments 8: The authors may add more detection articles for applications in various fields ( 3D vision technologies for a self-developed structural external crack damage recognition robot; Automation in Construction. Performance Metrics of an Intrusion Detection System Through Window Based Deep Learning Models; Journal of Data Science and Intelligent Systems.)
Response 8: Accept. We add some of the detection articles in the reference. The following papers are added to the reference:
- Hu K , Chen Z , Kang H ,et al. 3D vision technologies for a self-developed structural external crack damage recognition robot. Automation in construction, 2024(Mar.):159.
- Isiaka, F. Performance Metrics of an Intrusion Detection System Through Window-Based Deep Learning Models. Journal of Data Science and Intelligent Systems, 2023, 2(3), 174–180.
Round 2
Reviewer 1 Report
Comments and Suggestions for Authors
1.The main question addressed by the research is the autonomous stable tracking issue in electro-optical theodolite operating in closed-loop mode.
2.This research is partially improved and optimized based on the research of some researchers. It means that the research of the adaptive weighted fusion algorithm and a fusion tracking algorithm has certain novelty.
3.On the basis of the original part of the field, the innovation of this paper is to propose a multi-sensor adaptive weighted fusion algorithm and a fusion tracking algorithm based on a three-state transition model, which provides the reference direction for the field to solve the problem of photoelectric theodolite data fusion and autonomous stable tracking in the closed-loop tracking mode.
4.The author only uses simulation experiments to verify the feasibility of the proposed algorithm improvement and optimization. It should be verified in actual experiments
5.The conclusion presented by the author is basically consistent with the evidence and arguments presented. The problem of autonomous stable tracking of photoelectric theodolite in closed-loop mode is solved in some extent.
6.The references cited in the article are appropriate.
7. Quality of the figures should be improved. Size of the text in the figures is too small.
Author Response
Comments 1: The main question addressed by the research is the autonomous stable tracking issue in electro-optical theodolite operating in closed-loop mode.
Response 1: Accept. Thanks for your comments and appreciation on the paper.
Comments 2: This research is partially improved and optimized based on the research of some researchers. It means that the research of the adaptive weighted fusion algorithm and a fusion tracking algorithm has certain novelty.
Response 2: Accept. Thanks for your comments and appreciation on the paper.
Comments 3: On the basis of the original part of the field, the innovation of this paper is to propose a multi-sensor adaptive weighted fusion algorithm and a fusion tracking algorithm based on a three-state transition model, which provides the reference direction for the field to solve the problem of photoelectric theodolite data fusion and autonomous stable tracking in the closed-loop tracking mode.
Response 3: Accept. Thanks for your comments on the paper.
Comments 4: The author only uses simulation experiments to verify the feasibility of the proposed algorithm improvement and optimization. It should be verified in actual experiments
Response 4: Thank you for pointing this out. This paper shows indoor experiments, but various situations are simulated. For the actual environment, we can’t get the equipment data for the reason of secrecy. Basically, the indoor simulation experiment simulated all the actual situations. We add future work part in Chapter 4, the algorithm maybe improved for various application scenarios in the future work. {The modified can be found in line 566-567, page16}.
Comments 5: The conclusion presented by the author is basically consistent with the evidence and arguments presented. The problem of autonomous stable tracking of photoelectric theodolite in closed-loop mode is solved in some extent.
Response 5: Accept. Thank you very much for your comments.
Comments 6: The references cited in the article are appropriate.
Response 6: Accept. Thanks for your comments on this paper.
Comments 7: Quality of the figures should be improved. Size of the text in the figures is too small.
Response 7: Accept. Thank you for pointing this out. We have replaced the figures to improve the figure quality. See fig.1, fig.4, fig.7, fig.9, fig.11, fig.13, fig.15, fig.17 in the resubmitted manuscript.